# Differential protection by nicotinamide in a mouse model of glaucoma DBA/2J revealed by second-harmonic generation microscopy

**Vinessia Boodram, Hyungsik Lim**[¤]*

Department of Physics and Astronomy, Hunter College of the City University of New York, New York, NY, United States of America

¤ Current address: School of Optometry, Indiana University, Bloomington, IN, United States of America
* hl128@iu.edu

**Data Availability Statement:** All relevant data are within the paper and its supporting information files.

## Abstract

Glaucoma is a blinding disease where the retinal ganglion cells and their axons degenerate. Degradation of axonal microtubules is thought to play a critical role in the pathogenesis, but the mechanism is unknown. Here we investigate whether microtubule disruption in glaucoma can be alleviated by metabolic rescue. The integrity of axonal microtubules and the morphology of the retinal nerve fibers were evaluated by second-harmonic generation microscopy in a mouse model of glaucoma, DBA/2J, which received a dietary supplement of nicotinamide (NAM) for reducing metabolic stress. It was compared with control DBA/2J, which did not receive NAM, and non-glaucomatous DBA/2J-*Gpnmb+*. We found that the morphology of the retinal nerve fibers, but not axonal microtubules, are significantly protected by NAM. The decoupling is analogous to microtubule deficit, a glaucoma pathology in which axonal microtubules exhibit rapid degradation compared to the morphology of the retinal nerve fibers. Understanding microtubule deficit could provide insights into the divergent responses to NAM. From co-registered images of second-harmonic generation and immunofluorescence, it was determined that microtubule deficit was not due to a shortage of tubulins. Furthermore, microtubule deficit colocalized with the sectors in which the retinal ganglion cells were disconnected from the brain, suggesting that microtubule disruption is associated with axonal transport deficit in glaucoma. Together, our data suggests significant role axonal microtubules play in glaucomatous degeneration, offering a new opportunity for neuroprotection.

## Introduction

Glaucoma is a leading cause of blindness worldwide [1, 2], where nearly half of the retinal ganglion cells (RGCs) and their axons are irreversibly lost by the time of diagnosis. The risk factors include aging and high intraocular pressure (IOP), but the pathogenic mechanism is poorly understood hampering the prevention of vision loss. Axonal microtubules have been postulated to play a crucial role in the disease [3–5]. It has been demonstrated that in DBA/2J (D2)

**Funding:** This work was supported by funding from the National Institute of Health, EY033047 and GM140841 (H.L.). The funders had no role in study design, data collection and analysis, decision to publish, or preparation of the manuscript.

**Competing interests:** The authors have declared that no competing interests exist.

mice, a well-characterized model of inherited glaucoma [6–9], microtubules in RGC axons (i.e., the retinal nerve fibers) decay with age more rapidly than the axons themselves [10]. The degradation of microtubules would impair axonal transport depriving the RGCs of essential tropic and metabolic supports. One of the possible drivers of microtubule loss is dysfunctional cellular metabolism, compromising active regulation that stabilizes the microtubule assembly. Microtubule stability and metabolic equilibrium are co-dependent: Conversely, disruption of microtubules, which enable intracellular transport of mitochondria, could induce local energetic shortfalls in the distal RGC compartments. Lowering metabolic stress with metabolites, e.g., nicotinamide adenine dinucleotide (NAD) or its precursor nicotinamide (NAM), has been shown to delay or halt axonal degeneration [11] and protect the RGCs against glaucoma [12, 13]. It is plausible that the mechanism of neuroprotection by NAD/NAM involves axonal microtubules.

Here, using NAM as a paradigm, we investigate how axonal microtubules of the RGCs respond to metabolic rescue. Second-harmonic generation (SHG) microscopy was employed for measuring microtubules in the retinal nerve fibers, which, by virtue of the sensitivity to uniformly polarized microtubules [14–16], provides distinctive information regarding the cytoskeleton's integrity.

## Results

An overview of the experiment is shown in Fig 1(A). A group of D2 mice (N = 41 eyes, 11 eyes from 8 males and 30 eyes from 16 females) received a dietary supplement of NAM prophylactically from 6 months of age at a low dose of 550 mg/kg body weight per day, as previously described (12, 13). As a control, another group of D2 mice received a normal diet without NAM (N = 33 eyes, 13 eyes from 8 males and 20 eyes from 12 females). The strain-matched, homozygous DBA/2J-*Gpnmb+* mice (D2-*Gpnmb+*) were used as a non-glaucomatous control [17] (N = 19 eyes, 13 eyes from 8 males and 6 eyes from 4 females). At a desired age, the IOP was measured using a TonoLab tonometer [18] on three different dates prior to imaging, and the average was taken. Mosaics were acquired over a region around the optic nerve head (Fig 1 (B)). Image processing was carried out to normalize the SHG intensity and evaluate the thickness of the retinal nerve fiber bundles [10] (Fig 1(C)). The normalized SHG intensity divided by thickness, namely SHG density, provides a measure of the density of axonal microtubules at a microscopic resolution. The mean SHG density was evaluated over the retinal nerve fibers in the mosaic. Also, the thickness was integrated over the region to obtain the volume of the retinal nerve fibers.

### Morphology, but not microtubule density, of the retinal nerve fibers are protected by NAM

The effect of NAM diet was investigated on three properties, i.e., IOP, the volume and the mean SHG density of the retinal nerve fiber bundles (Fig 2 and S1 Table). D2 data contained the intrinsic variations of the strain and the experimental errors. The size of the latter can be estimated approximately from the variations of D2-*Gpnmb+* data. Overall, the variations of D2 were greater than those of D2-*Gpnmb+*. The age-dependent changes of the D2 eyes were analyzed using a multiple linear regression model containing three main effects that are known to influence the pathology (i.e., age, sex, and diet) as well as two interactions (summarized in Table 1). While the IOPs of D2-*Gpnmb+* mice remained stable, those of D2 without NAM increased with age, as expected. In D2 mice, the rate of IOP elevation with age varied depending on diet, which was significantly lower with NAM than without NAM ($p = .046$). Diminished IOP elevation has also been observed in the prior study but at a

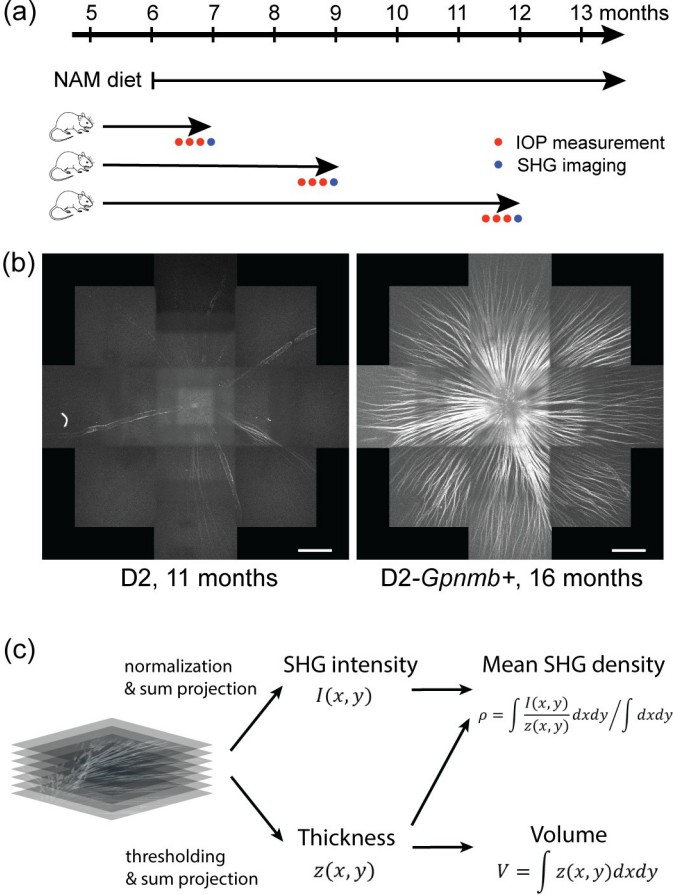

**Fig 1. Overview of the experiment.** (a) Timeline of NAM diet, IOP recording, and SHG imaging. (b) Representative SHG images of degenerated D2 versus healthy D2-*Gpnmb*+ retinas. Scale bars, 300 μm. (c) Image processing for evaluating the mean SHG density ρ and the volume *V*, representing 'microtubule density' and 'morphology', respectively.

higher dose of NAM [12, 13]. Interestingly, while the volume of the retinal nerve fibers responded to NAM exhibiting significantly lower age-dependent loss ($p$ = .049), the decay of the mean SHG density in D2 with NAM was not affected significantly ($p$ = .43). Thus, the benefit of NAM supplement was outstanding only for protecting the morphology, but not the microtubule density, of the retinal nerve fibers. Sex difference was not significant.

## Mean SHG density and volume are negatively correlated with IOP

In addition to age-dependence, we investigated how the mean SHG density and the volume depended on IOP, another major risk factor of glaucoma (Fig 3). In D2 mice without NAM, we found modest negative correlations at the level of significance between the IOP versus the mean SHG density (Pearson correlation coefficient r = -0.42, $p$ = .015) and versus the volume (r = -0.37, $p$ = .033), implying that the loss of axonal microtubules and RGC axons could be downstream effects of IOP elevation. Then we examined how the dependence on IOP was modified by NAM diet. Specifically, we asked whether the protective effect of NAM, instead of being a direct consequence of bolstered metabolism, was mediated entirely by the reduced IOP elevation, which would preserve the correlation coefficient. On the other hand, if NAM had an additional mode of action to prevent the loss of RGC axons, the correlation was

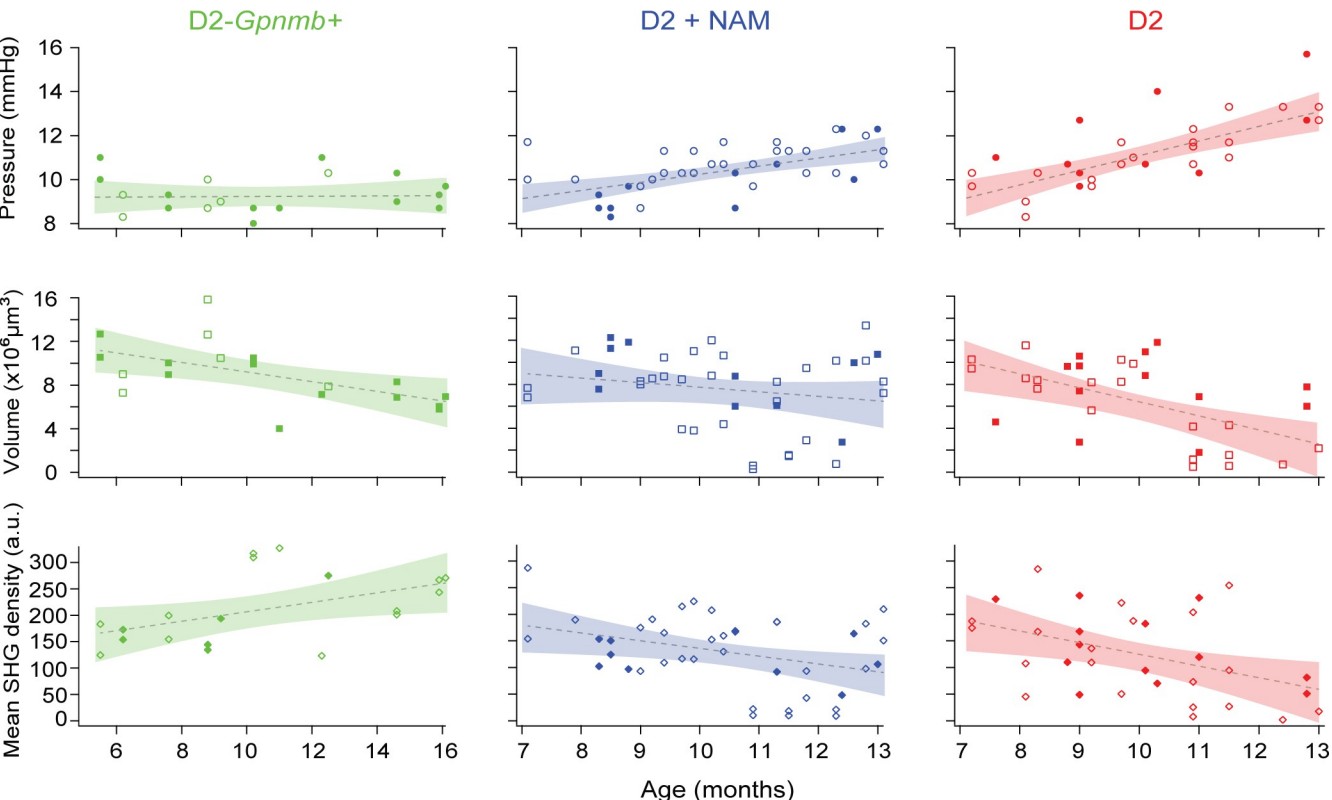

**Fig 2. Age-dependent effects of NAM dietary supplement on the D2 retinas.** IOP, the volume and the mean SHG density of the retinal nerve fibers versus age are shown for D2-*Gpnmb+*, D2 with and without NAM. Solid and open markers are male and female, respectively. Dashed lines are simple linear regressions with age and shaded bands are 95% confidence bounds.

expected to decrease as the IOP explains less of the variation. Unfortunately, the correlation coefficient between the volume and the IOP could not be determined with confidence for the group of D2 mice with NAM diet (r = -0.19, p = .24), leaving the inquiry unresolved.

## Microtubule deficit evolves in a NAM-dependent manner during glaucoma progression

Disruption of axonal microtubules before the loss of RGC axons culminates in a pathological state of glaucoma called *microtubule deficit*, where the retinal nerve fiber bundles are morphologically intact but contain less than normal amount of microtubules [10]. Based on the differential protection of the volume and the microtubule density of the retinal nerve fibers, it was anticipated that NAM diet will alter the evolution of microtubule deficit in glaucoma. To illustrate this, the mean SHG density and the volume were analyzed jointly in the same samples. Provided that the rate coefficients of the mean SHG density ρ and the volume V are given by α and β, respectively,

$$\alpha = -\frac{1}{\rho} \cdot \frac{d\rho}{dt} = -\frac{dlog\rho}{dt}, \ \beta = -\frac{1}{V} \cdot \frac{dV}{dt} = -\frac{dlogV}{dt}$$

, then the ratio of coefficients is obtained from the slope of a log-log plot.

$$\frac{\alpha}{\beta} = \frac{-dlog\rho/dt}{-dlogV/dt} = \frac{dlog\rho}{dlogV}$$

**Table 1. Summary of statistical analysis.** D2 mice with and without NAM (N = 74 eyes) were analyzed using a multiple linear regression model $y = \alpha + \beta_{age} \cdot Age + \beta_{sex} \cdot Sex + \beta_{diet} \cdot Diet + \beta_{int1} \cdot Age \cdot Sex + \beta_{int2} \cdot Age \cdot Diet + \varepsilon$.

| | Dependent variables | | |
|---|---|---|---|
| | IOP (mmHg) | Volume (x10$^6$ μm$^3$) | Mean SHG density (a. u.) |
| Age | 0.60*** | -1.50*** | -25.04*** |
| | (0.12) | (0.39) | (8.26) |
| Sex | -1.98 | -4.09 | -87.27 |
| | (1.56) | (5.08) | (106.52) |
| Diet | 2.14 | -7.80 | -66.32 |
| | (1.48) | (4.83) | (101.30) |
| Age·Sex | 0.17 | 0.58 | 9.04 |
| | (0.15) | (0.49) | (10.33) |
| Age·Diet | -0.29** | 0.94** | 7.86 |
| | (0.14) | (0.47) | (9.84) |
| Constant | 5.24*** | 20.76*** | 373.52*** |
| | (1.22) | (3.98) | (83.49) |
| Observations | 74 | 74 | 74 |
| R$^2$ | 0.47 | 0.25 | 0.18 |
| Adjusted R$^2$ | 0.43 | 0.20 | 0.12 |
| F Statistic | 12.03*** | 4.61*** | 2.93** |
| (df = 5; 68) | | | |

Note:

*$p < .1$

**$p < .05$

***$p < .01$

The ratio greater than unity ($\alpha/\beta > 1$) will be obtained in case of microtubule deficit, or if the volume is preserved better than microtubules. Overall, the D2 retinas without NAM had higher probabilities of microtubule deficit (Fig 4), reproducing the previous findings [10] with different experimental protocols and apparatus. It confirms microtubule deficit as a molecular marker of glaucoma. When the populations of D2 retinas with and without NAM diet were compared, the evolution of microtubule deficit was more appreciable in the D2 retinas with NAM than those without NAM, suggesting that the volume was protected in the progression of glaucoma better than axonal microtubules.

## Microtubule deficit is not due to paucity of tubulins

NAM exacerbates decoupling between axonal microtubules and morphology in glaucoma underlying microtubule deficit. Understanding microtubule deficit could allow an insight into the mechanism of divergent responses to NAM. The origin could be either a shortage of tubulin monomers, e.g., from inadequate expression or transport to the RGC axons, or a failure to stabilize the polymer, e.g., from dysfunctional regulation. To resolve these possibilities, we performed a co-registration of immunofluorescence and SHG images. The D2 retinas were fixed immediately after SHG imaging and double immunostained against two types of cytoskeletal protein, i.e., class III beta-tubulin (βIII) and phosphorylated neurofilament (pNF). βIII is a neuron-specific isoform of tubulin, and pNF is normally enriched in axons but accumulates in the soma of degenerating RGC's. To examine the microtubule integrity and the abundance of tubulins in the same regions, βIII immunofluorescence was co-registered with SHG images. A

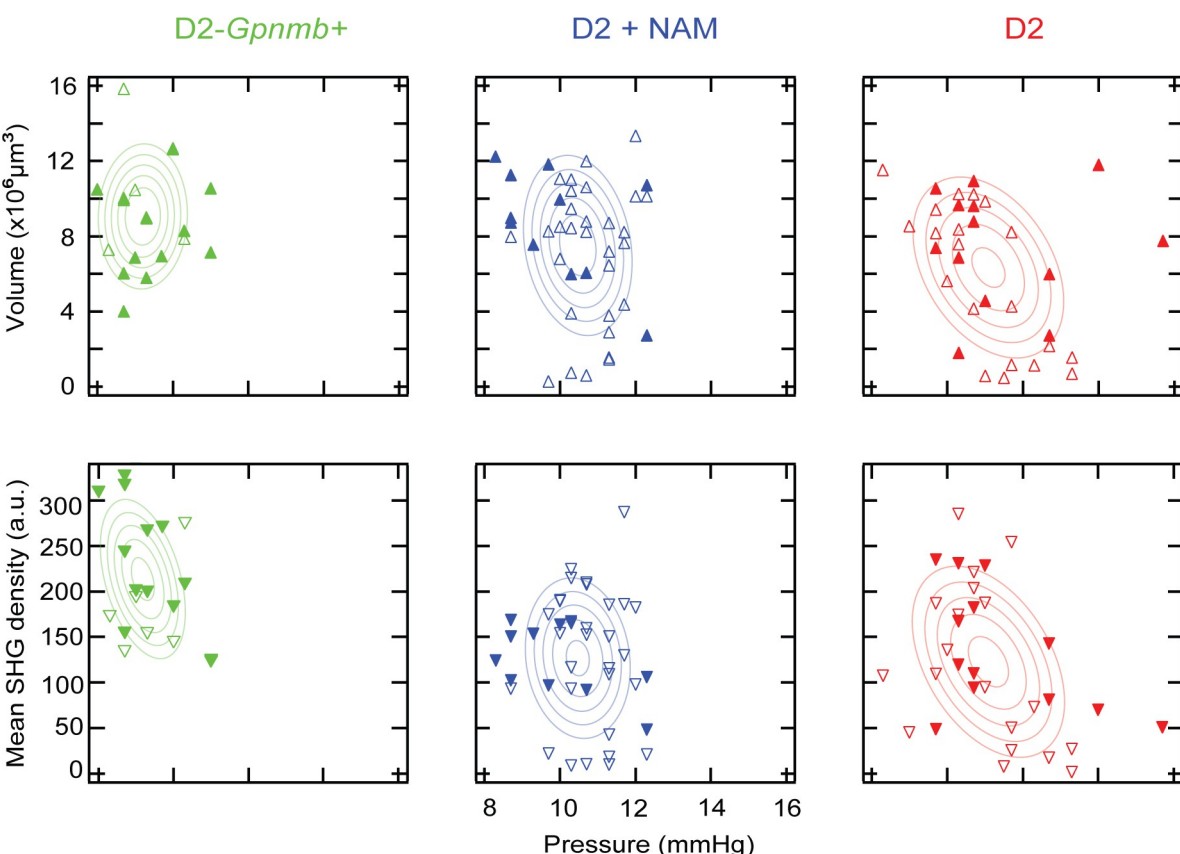

**Fig 3. Correlation between morphology/microtubules and IOP.** The correlation coefficients for D2 with and without NAM are compared to examine the effect of NAM diet. Solid and open markers are male and female, respectively, overlaid with the best fits to the bivariate normal distribution.

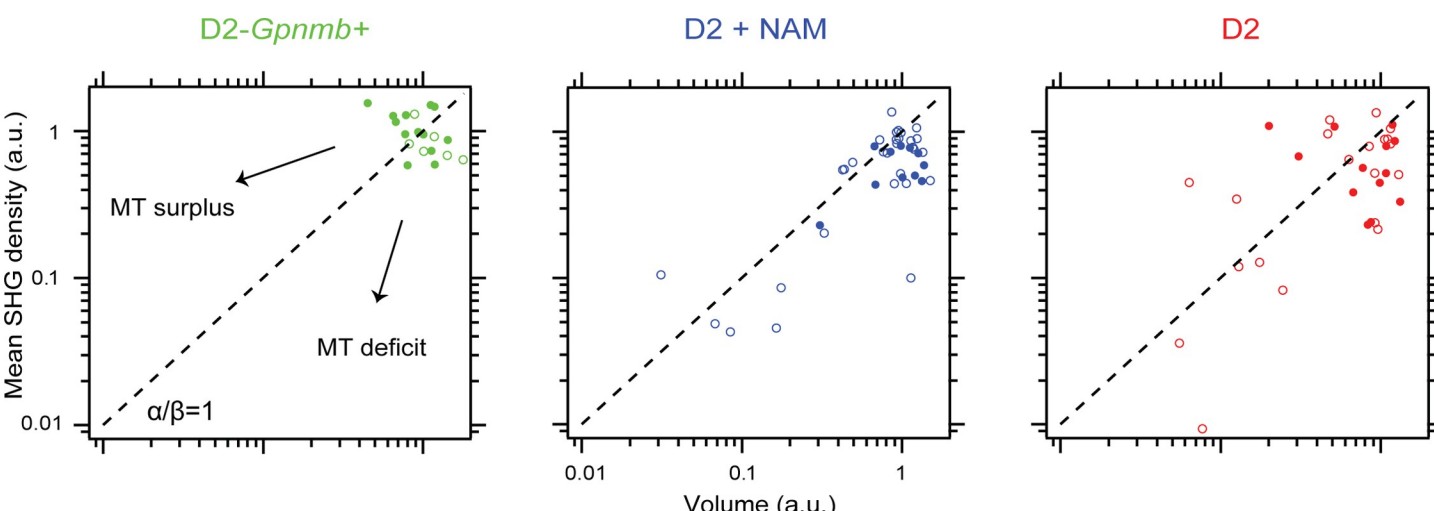

**Fig 4. NAM-dependent evolution of microtubule deficit.** The relative course of degeneration of volume and mean SHG density are different for three groups. The parameters are normalized to the average values of D2-*Gpnmb+* retinas. Solid and open markers are male and female, respectively. Dashed line, the unity slope. MT; microtubule.

total of twelve D2 retinas were evaluated, primarily focusing on the age range between 10 and 12 months. In the retinas exhibiting pronounced sectorial degeneration (N = 4), which is a characteristic in glaucoma pathology, the sectors of microtubule deficit always contained the morphologically intact retinal nerve fibers (Fig 5(A), between dashed lines). Also, it also showed high signals of βIII tubulins inside the sector, i.e., no substantial impairment in the expression or transport. It can be therefore concluded that microtubule deficit is not due to a lack of tubulins.

## Loss of axonal microtubules colocalizes with the RGC's disconnect from the brain

Axonal transport deficit is a milestone event in the progression of glaucoma, which is alleviated with NAM diet [12, 13]. The pathology could be caused by defects in the cytoskeleton or molecular motor proteins, whose functions rely on cellular energy. To elucidate the NAM's effect on axonal transport, first the relationship between axonal microtubules and transport deficit must be elucidated. The somatic accumulation of pNF, which can be detected with monoclonal antibody lacking nonspecific staining of soma (2F11), is interpreted to suggest that the RGC is disconnected from the brain, as has been validated by

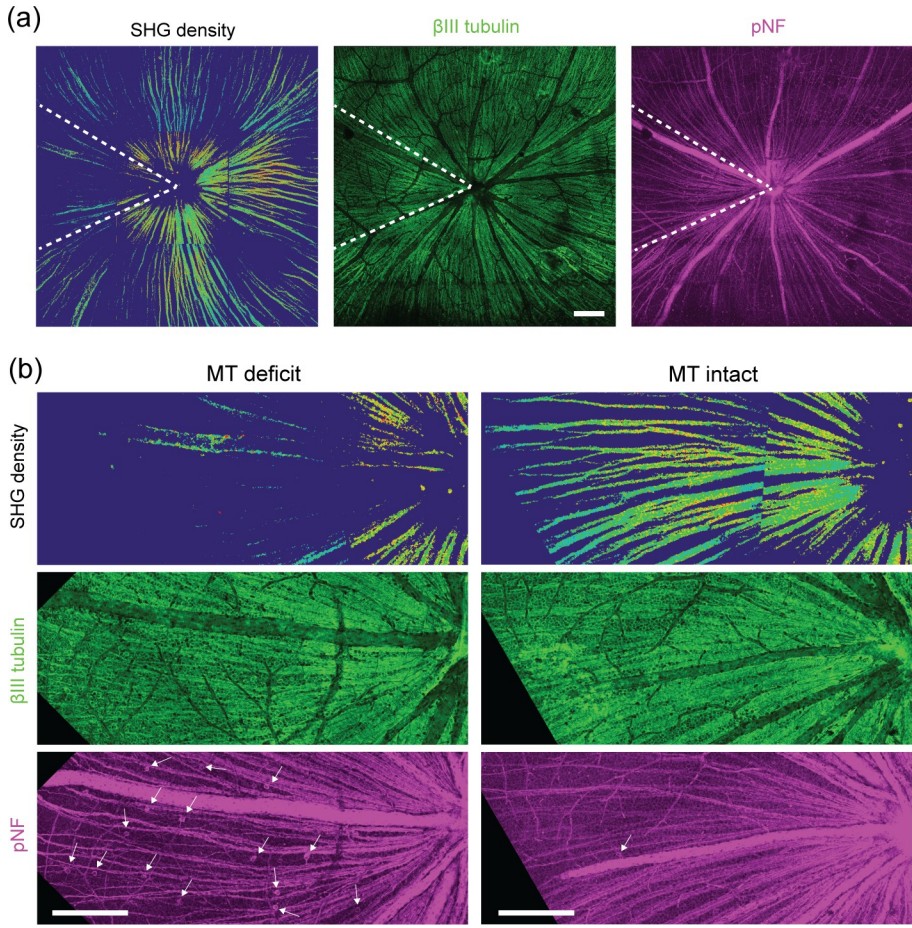

**Fig 5. Microtubule deficit versus the distribution of cytoskeletal proteins.** (a) Mosaics of a D2 retina (22 months age, female) by SHG and immunofluorescence against the cytoskeletal elements, βIII tubulin and pNF. (b) Comparing two sectors with deficient and intact microtubules. Arrows, pNF+ RGC somas. Scale bars, 200 μm.

retrograde tracing [19, 20]. To investigate the relationship between microtubule and transport deficits, we compared SHG density with the distribution of pNF. We found that pNF + somas represented a small subset of the RGC population (in the order of dozens in an area of approximately 1.8 mm diameter around the optic nerve head) (Fig 5(B)) and were absent in highly degenerated areas (S1 Fig). It could be because not all RGCs undergo this pathological fate during glaucomatous degeneration, and/or the redistribution of pNF could be short-lived with pNF+ RGC somas disappearing altogether upon the cells' death. Interestingly, pNF+ RGC somas were localized in sectors (Figs 5 and S1) and routinely coincided with that of low SHG density (Fig 5(B)). The result, together with intact βIII signals, suggests that the RGC's connection to the brain is likely to be severed at the time of microtubule disruption while RGC axons are intact, i.e., at the stage of microtubule deficit.

## Discussion

Taken together, our data supports a model in which NAM/NAD has several independent modes of action targeting distinct aspects of glaucoma pathogenesis. Considering that most cell types regulate diverse cellular processes in an energy-dependent manner, there could be non-RGC as well as RGC contributions in the protective effects of NAM/NAD. The IOP elevation has a component that responds to NAM, likely through the cells of the anterior segment, which is partially responsible for protecting the RGCs. It has been shown that NAM at a low dosage protects the RGCs without altering the IOP [12, 13], indicating that there is also IOP-independent protection. NAM reverses a variety of molecular changes in the RGCs that render them vulnerable to IOP elevation, e.g., mitochondrial dysfunction and synaptic elimination, which are likely to constitute the mechanism of IOP-independent protection. However, many aspects of IOP-independent protection are still obscure, e.g., how significant it is relative to IOP-dependent contribution. It is conceivable that the divergent responses to NAM are facilitated by an IOP-independent pathway. The morphology of RGC axons hinges on multiple conditions beside the integrity of microtubules. Contrary to axonal microtubules which might be primarily IOP-dependent, prevention of axon loss could be aided further by other IOP-independent factors that are bolstered metabolically by NAM. Detailed knowledge of distinct modes of action can improve the precision of NAM-based glaucoma therapy.

Differential protection by NAM also sheds critical insights into microtubule deficit. We hypothesized that metabolic stress, such as mitochondrial abnormalities or depletion of metabolites, might destabilize axonal microtubules to induce microtubule deficit. The hypothesis predicts that microtubule deficit will be mitigated by NAM supplement. However, our data do not support that microtubule disruption in glaucoma is strongly related to metabolic decline, leaving the pathogenic origin of microtubule deficit as a mystery. It is not a consequence of a shortage of tubulins in the RGC axon. Instead, microtubule deficit seems to be related to the elevated IOP, which exhibits a negative correlation with axonal microtubules. Regarding whether deteriorating axonal microtubules play a causative role in the eventual loss of RGCs, our multimodal data reveals spatially overlapping sectorial patterns of microtubule and axonal transport deficits. Correlated sectorial degenerations have allowed researchers to associate the loss of RGC axons with pathogenic insults at the optic nerve head [7, 21–23]. Similarly, the alignment of axonal transport and microtubule deficits suggests that the cytoskeletal breakdown might be an intermediate stage in the progressive RGC death. The proposed role of axonal microtubules offers a new opportunity for neuroprotection against glaucomatous degeneration. It also raises an important question regarding glaucoma therapy, i.e., whether the efficacy would be hampered without reversing the degradation of axonal microtubules.

Our discovery of persistent microtubule deficit with NAM seems inconsistent with a prior finding that anterograde axonal transport is recovered by NAM, as probed with cholera toxin β-subunit (CT-β) in the optic nerve head and the lateral geniculate nucleus [12]. It can be reconciled as follows: Although microtubule densities (i.e., mean SHG densities) are similarly low in NAM vs no-NAM D2 mice, the former has more abundant RGC axons thus could display more widespread CT-β signals. However, owing to persistent microtubule deficit, the CT-β signals are predicted to be much less intense in the individual cells of NAM D2 compared to those in the non-glaucomatous control (which has not been compared in the previous studies). Since microtubule deficit continues to worsen with age, axonal transport in NAM D2 mouse is likely to be compromised eventually at an age that microtubule density drops below a certain level, regardless of the integrity of RGC axons.

Our study has a few shortcomings. First, the protection of RGC axons by NAM/NAD could not be compared with that of RGC soma because RGC counts were not measured. The relative course of NAM/NAD effects in the distinct compartments of RGCs could provide additional insights into the mechanism of action. Second, our study did not include D2-*Gpnmb+* on NAM diet as a control group, which could have informed us whether the IOP-independent effect of NAM is specific to the glaucoma process or present also in the non-glaucomatous eyes. Third, our IOP measurement was insufficient for unraveling the mechanism quantitatively. Specifically, our measured IOP values of D2 mice were lower than those in the literature [24, 25]. Although the exact reason for inaccuracy is unclear (especially since the TonoLab device permits the user only basic controls), it seems due to a scaling error in the rebound tonometer, possibly incorrect software setting or mechanical deviations, given that the IOPs were underestimated uniformly across the ages thus displaying the anticipated overall trends for D2 as well as D2-*Gpnmb+* mice. Furthermore, the measurements were highly repeatable, yielding consistently close IOPs on three different dates (S1 Table). This type of error does not affect the overall conclusion of our study. Nonetheless, for achieving direct evidence of IOP elevation causing the pathology of RGCs, it would be highly desirable to improve the accuracy and the dynamic range of tonometer. It can be achieved by recruiting an independent IOP measurement. Cannulation and manometry is the gold standard of IOP measurement, which could provide the ground truths to calibrate TonoLab devices [26]. Although invasive, hence less suitable for repeat measurements, cannulation can avoid potential errors of TonoLab arising from age-dependent changes in the mechanical properties of the eye. Finally, quantitative analysis was confounded by the substantial unexplained variation of D2 mice. It could be beneficial to employ an alternative glaucoma model of IOP-induced rodents [27]. By isolating and regulating the IOP insult, it could resolve the relationship between the mitigation of IOP elevation and the metabolic rescue of RGCs by NAM/NAD. From the observed microtubule deficit's negative correlation with IOP, it is anticipated that the pathology is likely to be present also in other glaucoma models.

We have demonstrated that SHG imaging is well-suited for interrogating microtubule deficit. The information conveyed by the SHG signal is much distinguished from immunofluorescence against tubulin monomers, which is necessary but not sufficient for the integrity of microtubule assembly. Characterizing microtubule deficit across the whole retina, which is nontrivial with electron microscopy, is straightforward with SHG imaging. Being sensitive to the functional form of axonal microtubules, SHG provides an ideal readout for dissecting their mechanistic role in glaucoma pathogenesis.

## Materials and methods

### Animals

All procedures were approved by the Hunter College Institutional Animal Care and Use Committee (IACUC). DBA/2J (# 000671) and DBA/2J-*Gpnmb+* (# 007048) mice were purchased from The Jackson Laboratory and housed in the animal facility at Hunter College. DBA/2J mice were obtained at the ages of 8–12 weeks and housed until the desired age. Breeding pairs of DBA/2J-*Gpnmb+* were obtained at the age of around 4 weeks and the offsprings were raised to the desired age.

### Pharmacology

Nicotinamide was given at a low dose of 550 mg nicotinamide/kg body weight per day [12, 13] by adding to standard pelleted chow (2750 mg/kg in LabDiet 5001, based on an average 25-g mouse consuming 5 g of diet per day) (Bio-Serv).

### IOP measurement

IOP was measured using a TonoLab tonometer (Icare) following the manufacturer's manual. Mouse was awake (i.e., unanesthetized) and restricted during the measurement. Three measurements were performed on consecutive dates prior to SHG imaging. The time of measurement was standardized at a fixed hour during daytime to avoid the intraday fluctuations [28].

### Tissue preparation

The animal was deeply anesthetized with isoflurane and the first eye was enucleated. After the enucleation of the second eye, the animal was euthanized by $CO_2$ inhalation. The retinal flatmounts were prepared as previously described [10]. Briefly, an incision was made along the corneal limbus, the lens and sclera were removed, and radial cuts were made to relieve the curvature. The flat-mounted retina was transferred to a glass bottom dish (MatTek Corp.) and incubated at room temperature in the Ames' medium (A1420, Sigma-Aldrich) oxygenated with $95\%O_2/5\%CO_2$. Due to the lability of microtubules, SHG signals from the polymer last for a short period of time after enucleation ($\sim$ 2–3 hours). Therefore, for the accuracy of analysis, the samples were discarded if the flatmounts were not of high quality for 3D mosaics or too much time was elapsed for the preparation.

### SHG microscopy

An experimental setup for SHG microscopy was similar to the previous study [10, 15]. Briefly, 100-fs pulses at an 80-MHz repetition rate from a Ti:Sapphire laser (Chameleon Ultra, Coherent, Inc.) were used for the excitation. The output wavelength was 900 nm. The polarization state of excitation beam was controlled with half- and quarter-waveplates. A water-dipping microscope objective lens (HC FLUOTAR L 25x 0.95NA, Leica) was used to focus the excitation beam onto the sample. The average power was approximately 20 mW at the sample. The forward-propagating SHG from the sample was collected with an UV-transparent high-NA objective lens (UApo340 40× 1.35NA, Olympus), passed through a narrow-bandpass filter (<20-nm bandwidth) at a half of the excitation wavelength (400 nm), and then detected with a photomultiplier tube (PMT; H7422-40, Hamamatsu, Inc.). Images with 512×512 pixels were acquired, and the pixel dwell time was $\sim$ 3 μs. A region was imaged twice for orthogonal linear polarizations, which then were summed into a composite image. Z-stacks were acquired in a step of 2 μm. For creating mosaics, a total of 9 regions (742x742 $μm^2$ each) were imaged on and around the optic nerve head at 1-mm radius.

## Immunohistochemistry and confocal microscopy

Immunohistochemistry was performed similar to the prior studies [20, 23, 29]. After SHG imaging, the retinal flatmount was fixed with 4% paraformaldehyde for 20 minutes at room temperature. The sample was dehydrated sequentially in 25%, 50%, 75%, and 100% cold methanol for 15 min each and then permeabilized with dichloromethane for 2 hours. Then the retina was rehydrated in 75%, 50%, 25%, and 0% cold methanol for 15 min each. The sample was blocked in a buffer containing 5% normal serum and 0.3% Triton™ X-100 (Thermo Fisher Scientific, Inc.) for 3 hours. It was incubated in the primary antibody buffer at 4˚C for 3 days. Rabbit and mouse monoclonal antibodies against βIII tubulin (EP1569Y, Abcam, Inc.) and pNF (2F11, EMD MilliporeSigma), respectively, were used at 1:300 dilution. The sample was incubated in the secondary antibody buffer at 4˚C for 2 hours. Goat anti-rabbit and anti-mouse IgG antibodies conjugated with Alexa Fluor 594 and Alexa Fluor 647, respectively, were used (AB150080 and AB48389, respectively, Abcam, Inc.). The sample was mounted on a glass slide in Vectashield medium (Vector Laboratories, Inc.), and then imaged with a Leica TCS SP8 DLS confocal microscope using an oil-immersion objective lens (HC PL APO CS2 40× 1.3NA, Leica).

## Image analysis

Image processing was done using ImageJ [30] and MATLAB (MathWorks, Inc.). Mosaics were created using an ImageJ stitching plugin [31]. SHG normalization and thickness estimation was done as previously described [10]. Briefly, the composite SHG intensity was corrected for topography and normalized by dividing with the Fano factor. The thickness of the retinal nerve fiber was evaluated by means of context-free image segmentation, which was done through edge detection and histogram-based thresholding. The thresholded z-stack images were sum-projected and then multiplied with the z-step (2 μm) to obtain the thickness of the retinal nerve fibers. The SHG density was obtained by dividing the normalized SHG intensity by thickness. The volume of the retinal nerve fibers was evaluated by integrating the thickness over the region.

## Statistical analysis

Statistical analysis was performed using R [32]. The homogeneity of variance and normality were verified by inspecting the residual plots and the Q-Q plot. The collinearity of the explanatory variables was tested by the variance inflation factors. The IOP, the volume, and the mean SHG density were analyzed as the response variables with a multiple linear regression model. Three main effects of age, sex, and diet were considered. Of three possible interactions, the term between sex and diet was dropped on the ground that the effect of NAM is not known to be dependent on sex. Also, to aid model specification, the fit of multiple linear regression model was assessed by the adjusted R-squared. The alpha level for statistical significance was 0.05.

## Supporting information

**S1 Table. The measurement values of D2 and D2-*Gpnmb*+.**
(DOCX)

**S1 Fig. Microtubule deficit versus the distribution of cytoskeletal proteins.** (a) Mosaics of a D2 retina (12 months age, female) by SHG and immunofluorescence against the cytoskeletal elements, βIII tubulin and pNF. Scale bars, 200 μm. (b) Comparing two sectors in different

stages of RGC degeneration. Arrows, pNF+ RGC somas. Scale bars, 100 μm.
(TIF)

## Author Contributions

**Conceptualization:** Hyungsik Lim.

**Data curation:** Vinessia Boodram, Hyungsik Lim.

**Formal analysis:** Hyungsik Lim.

**Funding acquisition:** Hyungsik Lim.

**Investigation:** Vinessia Boodram, Hyungsik Lim.

**Supervision:** Hyungsik Lim.

**Writing – original draft:** Hyungsik Lim.

**Writing – review & editing:** Hyungsik Lim.

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
