## [Decision Letter · Decision Letter 0]

27 May 2024

PONE-D-24-15544Protective effects of nicotinamide in a mouse model of glaucoma DBA/2 studied by second-harmonic generation microscopyPLOS ONE

Dear Dr. Lim,

Thank you for submitting your manuscript to PLOS ONE. After careful consideration, we feel that it has merit but does not fully meet PLOS ONE’s publication criteria as it currently stands. Therefore, we invite you to submit a revised version of the manuscript that addresses the points raised during the review process.

We were able to obtain two reviews from people very respected in this kind of work. There were elements of excitement for the work, but as you'll see, they raise a significant number of issues with the current manuscript - all of which must be addressed. In keeping with the scope of PLoS One, I would emphasize that there is **no** weight applied regarding perceived impact. Thus, it is fine to point out caveats/weaknesses in the discussion, to refer to some aspects of the experiment as "negative" or as "interesting observations", to discuss aspects of the experimental design that might have been better designed knowing what is now known, to replace words like "indicates" with "suggests", and so forth. There **is** a requirement that "what is said matches what was done" and that nothing is over-stated. Thus, I believe with careful editing that a revised manuscript that addresses the Reviewers comments in the above context is possible (with no additional required experiments), but the editing will need to be extensive.  In addition to the comments of the two critiques, I would ask that a few sentences, or even a Figure of some sort, be added that help simply explain (and/or illustrate) how the measurements with second harmonics are related to the simplified terms of "morphology" vs "microtubules" that are used reiteratively. I think this would help make the paper more approachable to a broad readership.

We look forward to receiving your revised manuscript.

Kind regards,

Michael G Anderson, PhD

Academic Editor

PLOS ONE

Journal Requirements:

"This work was supported by funding from the National Institute of Health, EY033047 and GM121198 (H.L.)."

Reviewers' comments:

Reviewer's Responses to Questions

**Comments to the Author**

1. Is the manuscript technically sound, and do the data support the conclusions?

Reviewer #1: Partly

Reviewer #2: Partly

2. Has the statistical analysis been performed appropriately and rigorously? 

Reviewer #1: Yes

Reviewer #2: Yes

3. Have the authors made all data underlying the findings in their manuscript fully available?

Reviewer #1: Yes

Reviewer #2: No

4. Is the manuscript presented in an intelligible fashion and written in standard English?

Reviewer #1: Yes

Reviewer #2: Yes

5. Review Comments to the Author

Reviewer #1: This manuscript by Boodram and Lim explores microtubule disruption following elevated intraocular pressure in a glaucoma model. The Authors then utilize nicotinamide (the amide of vitamin b3 and a precursor to NAD) to perform a metabolic rescue. The main findings of the study is that glaucoma in the DBA/2J mouse results in microtubule loss and loss of morphology which is partly rescued by nicotinamide treatment. Although the paper is of interest and supports the growing literature on metabolic dysfunction in glaucoma, the data presented it not sufficient to support the claims. The following should be considered:

1. Low n and lack of power:

D2 glaucoma is high heterogenous (a benefit and caveat in the model). This means that typically 30-60 eyes are used in any one experiment. The data presented here is thin and highly variable. In fact, the Authors claim that this might be due to experimental error (!).

How can the Authors, or the Readers be convinced of the data?! The n is also strangely presented - is it n = images, n = eyes, n = mice? For example: "(N=19, 13 from 8 males and 6 from 4 females)" - what does this mean? Does it refer to the number of eyes and if so what was the criteria for choosing that eye?

Although the slopes look convincing, they are highly overlapping and even if the stats support the slope, it might be a statistically relevant, but not biologically relevant, finding. It should also be noted that the D2 mice have very low IOPs in these slopes. This is likely due to the low n, but this Reviewer also questions the use of the TonoLab on D2 mice (which has been shown to give erroneous readings unless done frequently, repeatedly, and on large numbers of mice.

2. Proper use of D2 mice.

It would be helpful to refer to DBA/2J mice and their WT counterparts using the correct nomenclature and abbreviations (e.g. D2). This will help with searching for these studies and indexing.

How do the Authors justify their staging of D2 glaucoma? D2 glaucoma is highly variable across animal facilities based on climate, diet, pathogens etc. More information is required. Preferably some levels of RGC counts or axon counts should be given and not just IOP in the matrix (which has a high level of flux).

How was the NAM diet made and administered? It is not sufficient to reference a paper from another group (it is barely acceptable to reference yourself rather than putting the methods down). Was NAM chow provided ad libitum? Did the Authors control the volume of the food or follow how much the mice were eating to make sure it was 550 mg/kg/d?

In the 'Animals' section it said that mice were bought in. Were they bought in and bred? Or bought in at the right ages? If so, how long were they acclimatized for? What was their diet before they were brought into the colony?

3. Other general comments:

PLOSONE is not a specialized journal. Both the intro and discussion are lacking in information. Why did you do these experiments? What is the background? What is the meaning? What is the literature that led you to your hypotheses?

Figure legends are too brief and should more accurately explain their subject matter. Figures would also benefit from legends.

Following NAM treatment microtubule dysfunction is not fully recovered. The Authors discuss axon transport loss, however, 2 independent groups have shown that NAM prevents loss of axon transport and recovers PERG (PMID: 32605122, 28209901), so what is the mechanism or the Author's understanding of these results?

Reviewer #2: The authors evaluate the effects of nicotinamide on microtubules in DBA/2 glaucoma. Samples sizes are good (maybe a little on the low side) and timepoints evaluated are appropriate for this model. The results are modest with small effects that appear independent of a reduction in IOP elevation making drawing conclusions of the effects of NAM on microtubules directly challenging. Overall, the authors need to more carefully consider the results and not be afraid to state negative finding as an important contribution to the field. Specific issues that also need to be addressed are:

Major comments:

Table 1: Individual measurements (e.g. IOP readings) should be shown, not just summary statistics.

Lines 80-85: IOP by tonolab versus IOP by cannulation. Worth stating differences in accuracy given differences in this study to previous reports. Also, no relationship between IOP and RGC loss has been observed before, likely due to the fact that a single measurement of IOP is not reflective of IOP insult in DBA mice. Please include discussion.

Lines 151-152: Conclusion is overstated. Evidence shown is that there is no impairment in expression or transport. Please modify.

Results do not always include consideration of effect of NAM treatment, particularly in the last couple of sections. Given the title states “Protective effects of nicotinamide…” – these sections should be modified.

Overall, the Discussion could do with a more thoughtful interpretation of the data particularly in relation to previous studies in multiple models – beyond the DBA model.

Related to the limited discussion – figure 6 is too simplistic with little consideration of other complexities in human glaucoma and animal models, including the DBA/2J model. E.g., compartmentalized RGC stress as a result of IOP elevation, aging, mitochondrial deficits, glial cell activation to name a few.

Minor comments:

If mice came from The Jackson Lab, please refer to them as DBA/2J.

Line 59-60: Experimental description talks about N – make sure this refers to eyes and not mice

Line 177: What does “NAM promotes the morphology of the retinal nerve fibers more significantly than microtubules…” Please clarify.

Although not powered necessarily, are there any trends in sex differences?

6. PLOS authors have the option to publish the peer review history of their article (what does this mean?). If published, this will include your full peer review and any attached files.

Reviewer #1: No

Reviewer #2: **Yes: **Gareth Howell

---

## [Author Response · Author response to Decision Letter 0]

16 Jun 2024

We are grateful for the thoughtful, expertly comments provided by the editor and reviewers. They have allowed us to identify areas of weakness in the previous version where our findings weren’t described adequately for the general readership. We revised the manuscript extensively with more detailed technical information to improve the rigor. We believe all the raised concerns are addressed. We received the following common criticisms from both reviewers:

1. IOP measurements with TonoLab

Both reviewers raised concerns about the IOP measurements. The IOP values of D2 mice we measured with TonoLab were lower than those in the literature. We used TonoLab according to the manufacturer’s instructions. Since the machine does not allow the user any control over calibration, other than choosing mouse or rat in the settings, the reason for the difference is unclear to us. Nonetheless, we concluded the measurements were reliable because 1) the measured IOPs on three different dates were consistently very close and 2) they displayed anticipated overall trends, such as D2 mice with age and NAM diet, as well as vs. D2-Gpnmb+. Individual IOP readings, which reviewer #2 asked about, are included as a Supplementary information in the revised version (S1 Table). Given the high repeatability/reproducibility, it seems unlikely that the underestimation was due to the low n (either number of samples or measurements), as reviewer #1 suspects. Rather, it could be a result of incorrect software setting or mechanical deviations of the rebound tonometer (which might be also behind the wide variability found in the literature). Conceivably, it is possible to determine, for a particular instrument that we used, the relationship between the actual and measured IOPs by performing a linear regression with the ground truths (e.g., by manometer). However, while the presumed scaling error does not affect the overall conclusion of our study, new errors that are harder to correct might arise from the extra steps of calibration. Considering these factors, the IOP values are presented as originally obtained with our TonoLab device, but a new paragraph is added to the Discussion regarding the accuracy of IOP measurement, as reviewer #2 recommended.

2. N, the number of eyes

Both reviewers pointed out the ambiguity of what N means. All N in the manuscript indicates eye samples, which we clarified in the revision. Reviewer #1 asked why the number of eyes was less than twice the number of animals and whether it was due to some selection. It is not the case. Not all eye samples yielded data because of the stringent conditions of SHG imaging. SHG signal from axonal microtubules lasts only for a few hours after enucleation, then it gradually decreases (presumably as the cytoskeleton depolymerizes in the absence of active regulation). No fixation protocol is yet known to preserve the signal. Consequently, there is a limited time window within which the imaging experiment must be completed. Furthermore, the quality of fresh, unfixed flatmount preparation is crucial for acquiring stacks of microscopic images across the large region. For the rigor of data, we discarded the retina samples that either took too long to prepare the flatmount or the full 3D volume could not be captured. Because of these demanding requirements, the success rate of SHG imaging was less than 100%. This information is added to the Methods. 

3. DBA/2J (D2): Nomenclature and Animal section

DBA/2J (or D2) are used to denote the strain, as both reviewers suggested. Reviewer #1 asked about the acclimation. DBA/2J mice were brought in from The Jackson Lab at the ages of 8-12 weeks and housed until the desired age. Breeding pairs of DBA/2J-Gpnmb+ were obtained at the age of around 4 weeks and the offsprings were raised to the desired age. The information is now included in the Methods.

4. Figure legends

Both reviewers commented that figure captions are insufficient. We made substantial improvements in the revision.

Other comments:

5. (Reviewer #1) D2 variability and experimental error

Our description of the experimental errors was misunderstood, i.e., “Overall, the variations of D2 data were greater than those of Gpnmb+ representing primarily the experimental errors.”. The sentence is rewritten to clarify this.

6. (Reviewer #1) Microtubule deficit vs. PERG and anterograde axonal transport of CT-β

Reviewer #1 noted that other labs have shown that NAM prevents loss of axon transport and recovers PERG (PMID: 32605122, 28209901). Assuming that PERG is more correlated with the integrity of RGC axons (i.e., morphology) than microtubule cytoskeleton, it is understandable that NAM diet improves PERG signal. We can also reconcile the recovered anterograde axonal transport with our finding, i.e., that the mean SHG density, which is proportional to the density of microtubules per axon caliber, is not significantly protected by NAM as follows: Between NAM vs. no-NAM D2 retinas, as compared in the previous studies (Williams et al.), there are more total microtubules in the former, despite similarly low densities of microtubules, just because there are more RGC axons. Provided that the spatial extent of CT-β signals (both in the ONH and LGN) is correlated with the total amount of microtubules rather than with the density, it is expected that the NAM D2 retina will display more widespread CT-β signals than the no-NAM D2 retina. Based on our data, however, individual RGCs of the NAM D2 retina still suffer microtubule deficit. So, the CT-β signals are predicted to be much less intense in individual cells of the NAM D2 retina than those of the non-glaucomatous control (which Williams et al. did not compare). Furthermore, our results show that microtubule deficit (i.e., low mean SHG density) continues to worsen with age, even with NAM diet. Consequently, we predict that axonal transport in the NAM D2 retina will be compromised eventually at an age when the microtubule density drops below a certain level, regardless of the integrity of RGC axons. We included this as a paragraph in the Discussion.

7. (Reviewer #1) Background for the hypothesis of the study

Reviewer #1 asked for an appropriate context leading to the present study. The question, in our opinion, is related to the editor’s comment about our insufficient explanation of "morphology" vs "microtubules". We addressed this by including Fig. 1(c), as suggested, and clarifying the concepts in the revised text. 

8. (Reviewer #1) Staging of D2 glaucoma

Reviewer #1 asked about the staging of D2 glaucoma. Unfortunately, our experimental design did not include the measurement of RGC counts. Only IOP and the properties of RGC axons, i.e., volume and mean SHG density, were measured as continuous independent variables, and their relationships were examined with age. The volume of the retinal nerve fibers can be considered an optical surrogate of RGC axon counts. We discuss this as one of the limitations of our study in the revision.

9. (Reviewer #1) NAM administration

Reviewer #1 questioned about NAM administration. NAM was added to the standard Rodent LabDiet 5001, the same diet the D2 mice received before switching to NAM-containing diet. For a dosage of 550 mg nicotinamide/kg body weight per day, we included 2750 mg nicotinamide/kg diet based on an average 25-g mouse consuming 5 g of diet per day. This information is added to the Methods.

10. (Reviewer #2) Last few sections of Results are not related to NAM treatment.

Reviewer #2 noted that the last two sections are not related to NAM treatment, which, considering that the title is “Protective effects of nicotinamide…”, should be modified. We agree with this critique. Those sections seek to investigate the details of microtubule deficit, or decoupling of morphology and microtubules, which was also evident in the response to NAM treatment. To address this comment, we changed the title to “Differential protection by nicotinamide…” and the sections are revised accordingly. 

11. (Reviewer #2) Limited discussion 

There were two related comments by Reviewer #2 regarding the limited discussion, i.e., that figure 6 is too simplistic and that the sentence “NAM promotes the morphology of the retinal nerve fibers more significantly than microtubules.” should be clarified. In response, we revised Fig 6 and the caption, and rewrote the first paragraph of Discussion entirely to enhance clarity.

Reviewer #2 encouraged us to include “a more thoughtful interpretation of the data particularly in relation to previous studies in multiple models – beyond the DBA model.”. In response, we added a new discussion comparing our finding of persistent microtubule deficit with the previously reported recovery of axonal transport by NAM. We also included a paragraph in the Discussion on how other glaucoma models beyond DBA/2 model could be beneficial for mechanistic studies of the NAM neuroprotection.

12. (Reviewer #2) Lines 151-152: Overstated conclusion regarding the origin of microtubule deficit.

The overstated conclusion is corrected.

---

## [Editor Report · Decision Letter 1]

21 Jun 2024

PONE-D-24-15544R1Differential protection by nicotinamide in a mouse model of glaucoma DBA/2J revealed by second-harmonic generation microscopyPLOS ONE

Dear Dr. Lim,

Thank you for submitting your manuscript to PLOS ONE. After careful consideration, we feel that it has merit but does not fully meet PLOS ONE’s publication criteria as it currently stands. Therefore, we invite you to submit a revised version of the manuscript that addresses the points raised during the review process.

As you will see below, there is a request to improve readability / clarity / cohesiveness prior to sending out for full re-review.

We look forward to receiving your revised manuscript.

Kind regards,

Michael G Anderson, PhD

Academic Editor

PLOS ONE

Additional Editor Comments:

Thank you for resubmitting your revised manuscript, which seems to have addressed several of the comments that were raised in initial review. However, prior to sending it back to the reviewers, I am asking that you first address a few important text issues that remain required. Between issues which appear to have been introduced during the extensive editing, and others remaining to be addressed, there is concern with readability that must be addressed. I hope handling these issues in this way might save time in the overall review process, and that they could be addressed with a modest amount of work – but they must be addressed to make the manuscript more readable and coherent.

1. The manuscript recurrently uses “morphology” to refer to the volume of the nerve fibers and “integrity” to refer to the SHG density. However, this distinction is not crisply defined and sometimes used in slightly different iterations that add confusion.

Please revisit how this central concept of the manuscript is addressed and strive to make it uniform and more readily appreciated. If the specific finding is that the NFL thickness is maintained by NAM, while the SHG density signal is not – it’s unclear why the additional terms are needed at all? (“Morphology” can refer to many things with respect to a retina or an axon).

Please revisit the descriptions for how volume was measured – it’s unclear if it’s based on IHC or inferred from the microscopy in general, if its synonymous with NFL thickness; or is a second measurement coming from the SHG imaging?

Throughout the manuscript (preferably at least in the Introduction and again in the Discussion) please integrate some language generic to an ophthalmic researcher describing what the two main measures likely relate to. Is “volume / morphology” the NFL thickness? If it comes from SHG, is the signal from the entire depth of the NFL, such that its decay would indicate NFL thinning? Is “density / integrity” from a single plane, or an average of each plane, or something else? Does it’s decay indicate that microtubules are lost or disorganized? Is the overall finding that NAM failing to protect SHG density an indication that NAM doesn’t protect the qualities of microtubules giving rise to SHG, perhaps that microtubules still become somehow structurally disorganized? If so, please use a sentence or two to restate the main finding and main implication plainly.

2. The term “microtubule deficit” is not defined until midway through the manuscript. If this phrase is going to be used as a noun to name a phenomenon, please define it early in the Introduction and again in the Methods.

3. In different places, citations 12 and 13 are used to support that NAM induces a change in IOP, and that it does not cause a change in IOP – please double check your intended meanings.

4. The phrase “…lower quantities than normal for the caliber” (L29, L128) is confusing – does this mean caliper of the individual axons or caliper of some other metric?

5. Issues 1-4 are present in the current iteration of the Abstract – with lines 26-30 particularly confusing. Please revisit to make sure that this portion of the manuscript can “stand alone” and does not use abbreviations or references to phenomenon that aren’t defined.

6. The Discussion raises several interesting points, but they don’t coalesce to make it clear what the primary model being proposed for glaucoma and NAM are with respect to microtubules, what the caveats/discordant data are, and which portions might best be called “speculative”.

---

## [Author Response · Author response to Decision Letter 1]

2 Jul 2024

We are grateful for the thoughtful, expertly comments provided by the editor and reviewers. They have allowed us to identify areas of weakness in the previous version where our findings weren’t described adequately for the general readership. We revised the manuscript extensively with more detailed technical information to improve the rigor. We believe all the raised concerns are addressed. We received the following common criticisms from both reviewers:

1. IOP measurements with TonoLab

Both reviewers raised concerns about the IOP measurements. The IOP values of D2 mice we measured with TonoLab were lower than those in the literature. We used TonoLab according to the manufacturer’s instructions. Since the machine does not allow the user any control over calibration, other than choosing mouse or rat in the settings, the reason for the difference is unclear to us. Nonetheless, we concluded the measurements were reliable because 1) the measured IOPs on three different dates were consistently very close and 2) they displayed anticipated overall trends, such as D2 mice with age and NAM diet, as well as vs. D2-Gpnmb+. Individual IOP readings, which reviewer #2 asked about, are included as a Supplementary information in the revised version (S1 Table). Given the high repeatability/reproducibility, it seems unlikely that the underestimation was due to the low n (either number of samples or measurements), as reviewer #1 suspects. Rather, it could be a result of incorrect software setting or mechanical deviations of the rebound tonometer (which might be also behind the wide variability found in the literature). Conceivably, it is possible to determine, for a particular instrument that we used, the relationship between the actual and measured IOPs by performing a linear regression with the ground truths (e.g., by manometer). However, while the presumed scaling error does not affect the overall conclusion of our study, new errors that are harder to correct might arise from the extra steps of calibration. Considering these factors, the IOP values are presented as originally obtained with our TonoLab device, but a new paragraph is added to the Discussion regarding the accuracy of IOP measurement, as reviewer #2 recommended.

2. N, the number of eyes

Both reviewers pointed out the ambiguity of what N means. All N in the manuscript indicates eye samples, which we clarified in the revision. Reviewer #1 asked why the number of eyes was less than twice the number of animals and whether it was due to some selection. It is not the case. Not all eye samples yielded data because of the stringent conditions of SHG imaging. SHG signal from axonal microtubules lasts only for a few hours after enucleation, then it gradually decreases (presumably as the cytoskeleton depolymerizes in the absence of active regulation). No fixation protocol is yet known to preserve the signal. Consequently, there is a limited time window within which the imaging experiment must be completed. Furthermore, the quality of fresh, unfixed flatmount preparation is crucial for acquiring stacks of microscopic images across the large region. For the rigor of data, we discarded the retina samples that either took too long to prepare the flatmount or the full 3D volume could not be captured. Because of these demanding requirements, the success rate of SHG imaging was less than 100%. This information is added to the Methods. 

3. DBA/2J (D2): Nomenclature and Animal section

DBA/2J (or D2) are used to denote the strain, as both reviewers suggested. Reviewer #1 asked about the acclimation. DBA/2J mice were brought in from The Jackson Lab at the ages of 8-12 weeks and housed until the desired age. Breeding pairs of DBA/2J-Gpnmb+ were obtained at the age of around 4 weeks and the offsprings were raised to the desired age. The information is now included in the Methods.

4. Figure legends

Both reviewers commented that figure captions are insufficient. We made substantial improvements in the revision.

Other comments:

5. (Reviewer #1) D2 variability and experimental error

Our description of the experimental errors was misunderstood, i.e., “Overall, the variations of D2 data were greater than those of Gpnmb+ representing primarily the experimental errors.”. The sentence is rewritten to clarify this.

6. (Reviewer #1) Microtubule deficit vs. PERG and anterograde axonal transport of CT-β

Reviewer #1 noted that other labs have shown that NAM prevents loss of axon transport and recovers PERG (PMID: 32605122, 28209901). Assuming that PERG is more correlated with the integrity of RGC axons (i.e., morphology) than microtubule cytoskeleton, it is understandable that NAM diet improves PERG signal. We can also reconcile the recovered anterograde axonal transport with our finding, i.e., that the mean SHG density, which is proportional to the density of microtubules per axon caliber, is not significantly protected by NAM as follows: Between NAM vs. no-NAM D2 retinas, as compared in the previous studies (Williams et al.), there are more total microtubules in the former, despite similarly low densities of microtubules, just because there are more RGC axons. Provided that the spatial extent of CT-β signals (both in the ONH and LGN) is correlated with the total amount of microtubules rather than with the density, it is expected that the NAM D2 retina will display more widespread CT-β signals than the no-NAM D2 retina. Based on our data, however, individual RGCs of the NAM D2 retina still suffer microtubule deficit. So, the CT-β signals are predicted to be much less intense in individual cells of the NAM D2 retina than those of the non-glaucomatous control (which Williams et al. did not compare). Furthermore, our results show that microtubule deficit (i.e., low mean SHG density) continues to worsen with age, even with NAM diet. Consequently, we predict that axonal transport in the NAM D2 retina will be compromised eventually at an age when the microtubule density drops below a certain level, regardless of the integrity of RGC axons. We included this as a paragraph in the Discussion.

7. (Reviewer #1) Background for the hypothesis of the study

Reviewer #1 asked for an appropriate context leading to the present study. The question, in our opinion, is related to the editor’s comment about our insufficient explanation of "morphology" vs "microtubules". We addressed this by including Fig. 1(c), as suggested, and clarifying the concepts in the revised text. 

8. (Reviewer #1) Staging of D2 glaucoma

Reviewer #1 asked about the staging of D2 glaucoma. Unfortunately, our experimental design did not include the measurement of RGC counts. Only IOP and the properties of RGC axons, i.e., volume and mean SHG density, were measured as continuous independent variables, and their relationships were examined with age. The volume of the retinal nerve fibers can be considered an optical surrogate of RGC axon counts. We discuss this as one of the limitations of our study in the revision.

9. (Reviewer #1) NAM administration

Reviewer #1 questioned about NAM administration. NAM was added to the standard Rodent LabDiet 5001, the same diet the D2 mice received before switching to NAM-containing diet. For a dosage of 550 mg nicotinamide/kg body weight per day, we included 2750 mg nicotinamide/kg diet based on an average 25-g mouse consuming 5 g of diet per day. This information is added to the Methods.

10. (Reviewer #2) Last few sections of Results are not related to NAM treatment.

Reviewer #2 noted that the last two sections are not related to NAM treatment, which, considering that the title is “Protective effects of nicotinamide…”, should be modified. We agree with this critique. Those sections seek to investigate the details of microtubule deficit, or decoupling of morphology and microtubules, which was also evident in the response to NAM treatment. To address this comment, we changed the title to “Differential protection by nicotinamide…” and the sections are revised accordingly. 

11. (Reviewer #2) Limited discussion 

There were two related comments by Reviewer #2 regarding the limited discussion, i.e., that figure 6 is too simplistic and that the sentence “NAM promotes the morphology of the retinal nerve fibers more significantly than microtubules.” should be clarified. In response, we revised Fig 6 and the caption, and rewrote the first paragraph of Discussion entirely to enhance clarity.

Reviewer #2 encouraged us to include “a more thoughtful interpretation of the data particularly in relation to previous studies in multiple models – beyond the DBA model.”. In response, we added a new discussion comparing our finding of persistent microtubule deficit with the previously reported recovery of axonal transport by NAM. We also included a paragraph in the Discussion on how other glaucoma models beyond DBA/2 model could be beneficial for mechanistic studies of the NAM neuroprotection.

12. (Reviewer #2) Lines 151-152: Overstated conclusion regarding the origin of microtubule deficit.

The overstated conclusion is corrected.

Additional Editor comments for improving the readability:

13. NFL thickness versus morphology

The Editor asks why the term ‘morphology’ is needed for indicating the volume of the retinal nerve fibers if it is synonymous with the ‘NFL thickness’. It would be a misrepresentation to replace morphology/volume with NFL thickness. The latter is commonly used in the ophthalmic field because ophthalmic imaging modalities (e.g., OCT) have inadequate lateral resolutions due to the low NA of the human eye. As a result, they cannot quantify precisely the retinal nerve fibers in the later dimensions; thus, NFL thickness is used as a surrogate measure of morphology. A major difference is that the volume, as we evaluated here, measures the lateral loss of the retinal nerve fibers as well as the axial loss, i.e., the NFL thinning. 

The volume was estimated from SHG images, as Fig 1(c) shows. The morphology of the retinal nerve fibers cannot be measured accurately by IHC due to tissue distortions (such as shrinking) occurring during the preparation.

As SHG intensity per thickness, SHG density probes specifically the changes in the microtubules regardless of the thickness.

The manuscript is revised to clarify these points. Whenever applicable, we tried to refer to the volume and SHG density, rather than morphology and microtubules, respectively. 

14. The definition of “microtubule deficit” 

The Editor asks to define the term early on. When the study was conceived, we were agnostic about the differential protection of NAM. In fact, our hypothesis was that both microtubules and morphology would be rescued. The connection to an unrelated phenomenon of microtubule deficit became relevant only after the data showed that the two properties responded differently to NAM treatment. So, we think it is important for the logical progression that the first mention of microtubule deficit appears after the result of divergence is presented. However, we agree that the term must be defined appropriately in Abstract so that it can stand alone. The manuscript is revised accordingly.

15. “…lower quantities than normal for the caliber” (L29, L128) is confusing.

Microtubule deficit is explained/defined with more clarity.

16. Citations 12 and 13 on NAM vs. IOP

NAM can induce changes of IOP or not depending on the dosage, according to the work cited in 12 and 13. The manuscript is revised to explain this prior art.

17. Unclear model in the Discussion

The model is to explain how IOP-independent pathway of RGC protection, which has been shown previously, might underlie the divergent response to NAM that we observed. The manuscript is revised to make this point lucid.

---

## [Decision Letter · Decision Letter 2]

24 Jul 2024

PONE-D-24-15544R2Differential protection by nicotinamide in a mouse model of glaucoma DBA/2J revealed by second-harmonic generation microscopyPLOS ONE

Dear Dr. Lim,

Thank you for submitting your manuscript to PLOS ONE. After careful consideration, we feel that it has merit but does not fully meet PLOS ONE’s publication criteria as it currently stands. Therefore, we invite you to submit a revised version of the manuscript that addresses the points raised during the review process. 

Your revised manuscript has addressed most points raised in initial review.  However, neither Reviewer yet recommended “Accept”. In considering their remaining concerns (expressed partially in the comments to the Authors, and partially in comments to the Editor) and balancing them with the scope of PLoS One, I ask you to address a few lingering minor points. Note that most of these are readily achievable by modest text changes in the Discussion.

From Reviewer 1. Note that the Reviewer indicated concerns with the technical soundness and statistical analyses of the manuscript. In keeping with the scope of PLoS One, two minor changes would likely address these concerns:

In the Discussion of caveats, I suggest that you add text pointing out that Gpnmb+ mice on NAM diet were not included as a control group but could have been informative.In the Results and/or Discussion (and Methods), please add a post hoc power analysis for at least one of the phenotypes and discuss the power briefly in the Discussion. Or, if the analyses are still in a discovery phase in which meaningful power analyses might be premature, please expand on what was learned from the current experiment that might guide future quantitative studies and point out the uncertain power of the current experiments in the caveats.

From Reviewer 2:

Regarding sufficiency of the presentation on caveats of rebound tonometry. I suggest that in the Discussion of IOP and the caveats associated with the rebound tonometer, please also note that cannulation (which some labs experienced with D2 mice consider a gold-standard for this strain) might also have helped achieve direct (or additional) evidence of IOP elevation across multiple ages.Regarding Figure 6. I agree with the Reviewer that the simplistic nature of this Figure has a modest impact on the manuscript, but as an online journal space is not particularly limiting and leave it to your discretion whether you prefer to keep or drop this Figure.Regarding IOP-independent mechanisms. Despite uncertainty in the mechanism, Williams has published on this topic and it would be appropriate to add a sentence or two and some citations.

Please submit your revised manuscript within Sep 07 2024 11:59PM. If you will need more time than this to complete your revisions, please reply to this message or contact the journal office at plosone@plos.org. Please include the following items when submitting your revised manuscript:A rebuttal letter that responds to each point raised by the academic editor and reviewer(s). You should upload this letter as a separate file labeled 'Response to Reviewers'.A marked-up copy of your manuscript that highlights changes made to the original version. You should upload this as a separate file labeled 'Revised Manuscript with Track Changes'.An unmarked version of your revised paper without tracked changes. You should upload this as a separate file labeled 'Manuscript'.If applicable, we recommend that you deposit your laboratory protocols in protocols.io to enhance the reproducibility of your results. Protocols.io assigns your protocol its own identifier (DOI) so that it can be cited independently in the future. For instructions see: https://journals.plos.org/plosone/s/submission-guidelines#loc-laboratory-protocols. Additionally, PLOS ONE offers an option for publishing peer-reviewed Lab Protocol articles, which describe protocols hosted on protocols.io. Read more information on sharing protocols at https://plos.org/protocols?utm_medium=editorial-email&utm_source=authorletters&utm_campaign=protocols.

We look forward to receiving your revised manuscript.

Kind regards,

Michael G Anderson, PhD

Academic Editor

PLOS ONE

Journal Requirements:

Additional Editor Comments:

In addition to my suggestions found above, please note that the Figures use "DBA" as a label - which should be corrected to either the full strain name (DBA/2J) or its standard abbreviation (D2).

Reviewers' comments:

Reviewer's Responses to Questions

**Comments to the Author**

1. If the authors have adequately addressed your comments raised in a previous round of review and you feel that this manuscript is now acceptable for publication, you may indicate that here to bypass the “Comments to the Author” section, enter your conflict of interest statement in the “Confidential to Editor” section, and submit your "Accept" recommendation.

Reviewer #1: All comments have been addressed

Reviewer #2: (No Response)

2. Is the manuscript technically sound, and do the data support the conclusions?

Reviewer #1: Partly

Reviewer #2: Yes

3. Has the statistical analysis been performed appropriately and rigorously? 

Reviewer #1: No

Reviewer #2: Yes

4. Have the authors made all data underlying the findings in their manuscript fully available?

Reviewer #1: Yes

Reviewer #2: Yes

5. Is the manuscript presented in an intelligible fashion and written in standard English?

Reviewer #1: Yes

Reviewer #2: Yes

6. Review Comments to the Author

Reviewer #1: Please see comments to the Editor. There are still outstanding issues that need to be addressed. Otherwise the manuscript has been significantly improved.

Reviewer #2: The authors have done a good job of responding the majority of the comments. However, a couple still remain.

1. Please state that IOP by cannulation is the gold standard for the DBA/2J model and errors in tonolab can arise due to changes to the eyes with age.

1. I do not think Figure 6 is necessary and should be removed.

2. The discussion about the IOP-independent mechanisms should fleshed out a little further, possibly simply with further reference to the Williams et al Science paper indicating protection at the level of mitochondrial abnormalities.

7. PLOS authors have the option to publish the peer review history of their article (what does this mean?). If published, this will include your full peer review and any attached files.

Reviewer #1: **Yes: **Pete A Williams

Reviewer #2: **Yes: **Gareth Howell

---

## [Author Response · Author response to Decision Letter 2]

1 Aug 2024

We thank the reviewers for their thoughtful comments. We made every effort to understand the concerns and revised the manuscript to address all of them as summarized below. 

(Reviewer 1) I suggest that you add text pointing out that Gpnmb+ mice on NAM diet were not included as a control group but could have been informative.

The text is included in the Discussion, as suggested. In hindsight, the control group could have informed us whether the IOP-independent effect of NAM is specific to the glaucoma process or also present in the non-glaucomatous eyes.

(Reviewer 1) Please add a post hoc power analysis for at least one of the phenotypes and discuss the power briefly in the Discussion. Or, if the analyses are still in a discovery phase in which meaningful power analyses might be premature, please expand on what was learned from the current experiment that might guide future quantitative studies and point out the uncertain power of the current experiments in the caveats.

Admittedly, we were not aware of the concept of post hoc power analysis prior to this comment. We found a fair amount of literature and online discussions on the topic, especially whether it is a meaningful analysis. Focusing on our study, we understand the question in hand is this: Given that we cannot reject the assumption that NAM diet does not have an effect on the mean SHG density (i.e., the null hypothesis), is there a sample size that we can calculate with which the effect could have been observed? The underlying reasoning is that the effect was not detected probably because it was too small and will be detectable with a larger sample size (or statistical power). The answer to the question is, yes, there can be such a sample size and, no, it cannot be calculated based on the current data. The explanation why such an estimation is flawed/misleading can be found in numerous papers (e.g., PMID: 11310512, 31552383, 29994928). In a nutshell, the estimated post hoc power is based on the variance of sample whereas the prospective power is on the variance of population. Because the estimated effect size is noisy, the power necessary to detect it (i.e., the post hoc power) is not informative. It has been proposed to present confidence intervals as an alternative (PMID: 11310512). This information is already in Fig.2, where the 95% confidence intervals are depicted around the slope of age-dependence for the two cases of diet.

(Reviewer 2) Please state that IOP by cannulation is the gold standard for the DBA/2J model and errors in tonolab can arise due to changes to the eyes with age. 

The statement is added to the Discussion, as suggested. 

(Reviewer 2) The discussion about the IOP-independent mechanisms should be fleshed out a little further, possibly simply with further reference to the Williams et al Science paper indicating protection at the level of mitochondrial abnormalities.

In response, we included a description of molecular changes reversed by NAM treatment, as shown in Williams et al., and the implications in the IOP-independent mechanism.

(Reviewer 2) I do not think Figure 6 is necessary and should be removed.

Figure 6 is removed, as suggested.

---

## [Editor Report · Decision Letter 3]

13 Aug 2024

Differential protection by nicotinamide in a mouse model of glaucoma DBA/2J revealed by second-harmonic generation microscopy

PONE-D-24-15544R3

Dear Dr. Lim,

We’re pleased to inform you that your manuscript has been judged scientifically suitable for publication and will be formally accepted for publication once it meets all outstanding technical requirements.

Kind regards,

Michael G Anderson, PhD

Academic Editor

PLOS ONE

Additional Editor Comments (optional):

Thank you for the final round of revisions. Sorry for the delay in accepting them as I was out of the country with no accessibility the past week.

I think the current manuscript has much solid data that the field will find meaningful. I would share that one of the reviewers was recurrently concerned with statistical power, which was minimally resolved, but might be helpful for your team to bear in mind as a concern for future work. For example, you might start incorporating and reporting pre-emptive power analyses. You could also report a calculation for a meaningful sample size if another group wanted to attempt to replicate a study - which the variability data from your study should be able to inform.
---

## [Editor Report · Acceptance letter]

30 Aug 2024

PONE-D-24-15544R3 

PLOS ONE

Dear Dr. Lim, 

I'm pleased to inform you that your manuscript has been deemed suitable for publication in PLOS ONE. Congratulations! Your manuscript is now being handed over to our production team.

Kind regards, 

on behalf of

Dr. Michael G Anderson 

Academic Editor

PLOS ONE